# GAN Quality Index (GQI) by GAN-induced Classifier

**Yuancheng Ye & Yingli Tian**
The City College and the Graduate Center
The City University of New York
yye@gradcenter.cuny.edu
ytian@ccny.cuny.edu

**Yue Wu**
Department of Electrical & Computer Engineering
Northeastern University
yuewu@ece.neu.edu

**Lijuan Wang, Yinpeng Chen, Zicheng Liu & Zhengyou Zhang**
Microsoft Research
{lijuanw, yiche, zliu, zhang}@microsoft.com

## Abstract

We propose an objective measure, called GAN Quality Index ($GQI$), to evaluate GANs. The idea is to train a GAN-induced classifier from the GAN generated data and use its accuracy on a real test set as a metric to measure how well the GAN model distribution matches the real data distribution. Unlike most existing quantitative measurements of GANs, which only reflect partial characteristics of generation distribution, the accuracy of the GAN-induced classifier can be used to derive a simple yet sufficient index to measure how well the generation distribution matches the true data distribution. We demonstrate the effectiveness of $GQI$ on CIFAR-100, Flower-102, and MS-Celeb-1M which contains 10,000 classes.

## 1 Introduction

Generative adversarial networks (GANs) (Goodfellow et al. (2014)) are a framework for producing a generative model by way of a two-player minimax game. Theoretically GANs learned from a real dataset are supposed to generate data with the same distribution as the real dataset, but it is still an open problem how to objectively evaluate a GAN to see if the generated data have the same distribution as that of the real data. So far, people have been relying on visual inspection to evaluate the quality of the generated images. Aside from being subjective, visual inspection cannot provide reliable information about the data distribution. In this paper, we propose to use classification as a metric to measure the performance of GANs.

Given a real dataset with $N$ classes, one can easily train a classifier $C_{real}$ and obtain its accuracy on a test set. Suppose we train a GAN on this dataset and assume we can obtain the labels of the generated data (later in Section 2, we will discuss how to obtain labels for those GANs which do not generate labels). One can also train a classifier $C_{GAN}$, called the GAN-induced classifier, with the generated data. If the generated data have the similar distribution as the real data, $C_{GAN}$ should have the

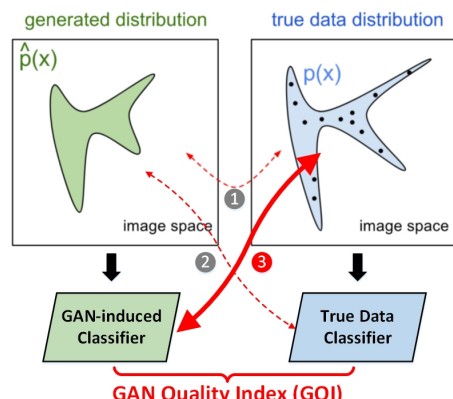

Figure 1: Different ways to evaluate GANs where the arrow labeled "3" is our newly proposed method based on the GAN-induced classifier.

similar performance as $C_{real}$ when applied to the real test data. Therefore, we believe that the classification capability of the GAN-induced classifier can be used as an objective measure to evaluate GANs.

There are few prior quantitative measures to evaluate GANs including using Gaussian Parzen windows (Goodfellow et al. (2014)), Generative Adversarial Metric (GAM) (Im et al. (2016)), Inception score (Salimans et al. (2016)), and Parzen window estimation (Lucas Theis (2015)). The existing methods can be roughly cateogrized into two categories depending on whether they use classification or not as shown in Figure 1. Arrow "1" includes those techniques that directly compare the distributions of the real data with the GAN data. Arrow "2" includes the method proposed by (Santurkar et al. (2017)) that trains a classifier on real data which is applied to the GAN data to generate the label distribution. Inception score can be thought of as belonging to "2" since it trains a classifier on ImageNet which is real data (even though not on the data used to train the GAN). Our method belongs to Arrow "3" that uses GAN data to train a classifier and applies it to real test data. The ratio between the accuracy of the GAN-induced classifier and the real-data trained classifier is used as the quality measure. We apply this measure to a number of representative GAN frameworks including unsupervised (GANs Goodfellow et al. (2014) and WGANs Arjovsky et al. (2017)), semi-supervised (SGANs Odena (2016); Salimans et al. (2016)), and supervised (CGANs Mehdi Mirza (2014)).

## 2 GAN QUALITY INDEX

Given a real dataset with $N$ classes, suppose a GAN is trained and its Generator is denoted as $G$. Meanwhile, a classifier is also trained on the real data, and we denote it as $C_{real}$. With enough randomized inputs in latent space, the Generator $G$ can generate many images, for example in our case one million images are generated for the MS-Celeb-1M task. The generated images are then fed into the pre-trained classifier $C_{real}$ to obtain the pseudo labels. A pre-set threshold is applied here to filter out the generated images with low pseudo label scores in order to discard the images with low quality or belonging to none of the existing $N$ classes. After these steps, we obtain a generated dataset with $N'$-class pseudo labels, and $m_i$ generated samples in the $i$-th class ($1 \leq i \leq N' \leq N$). The generated dataset with $N'$ class pseudo labels is used to train an independent classifier $C_{GAN}$, called the GAN-induced classifier. Both $C_{GAN}$ and $C_{real}$ are evaluated by the standard Top-1 accuracy on the same real test set. Let $ACC(C_{GAN})$ and $ACC(C_{real})$ denote the Top-1 accuracies of $C_{GAN}$ and $C_{real}$, respectively, the GAN Quality Index of $G$ is defined as

$$GQI = \lfloor \frac{ACC(C_{GAN})}{ACC(C_{real})} * 100 \rfloor. \tag{1}$$

$GQI$ is an integer in the range of 0 to 100 that indicates how close the accuracy of $ACC(C_{GAN})$ is to that of $ACC(C_{real})$. If $GQI = 100$, it means $ACC(C_{GAN})$ has the same accuracy as $ACC(C_{real})$. The higher the $GQI$, the closer the accuracy of $C_{GAN}$ to $C_{real}$ which indicates that the GAN distribution better matches the real data distribution.

## 3 EXPERIMENTS

**Datasets:** We have performed experiments on three public datasets: **CIFAR-100** (Krizhevsky (2009)), **Flower-102** (Nilsback & Zisserman (2008)), and **MS-Celeb-1M** (Guo et al. (2016)). CIFAR-100 dataset consists of 60,000 32x32 color images in 100 classes containing 600 images each. There are 500 training images and 100 test images per class. Flower-102 consists of 102 flower categories. The original split of this dataset has 1,020 images in the training set and 6,149 images in the testset. To use more training images for the CNN model, we take the larger set of 6,149 images for training and test on the smaller set of 1,020 images. MS-Celeb-1M-Base has 20,000 classes and a total of 1.2 million aligned face images, which is a smaller yet nearly noise-free version of MS-Celeb-1M dataset. We randomly choose 10,000 classes, and for each person, 80% with up to 30 images are randomly selected as the training images and the rest are used for testing.

**Network Training:** We use a 32-layer ResNet (He et al. (2015)) to train the real-data classifier $C_{real}$ for CIFAR-100. For Flower-102, an 18-layer ResNet is fine-tuned from a model that is trained with IMAGENET ILSVRC 2012 (Deng et al. (2009)) data. For MS-Celeb-1M, a 34-layer ResNet is used.

In all the GANs models, the network structure for the Generator $G$ is the same as the one in DC-GANs (Radford et al. (2015)). The Discriminator $D$ network of GANs, CGANs, and WGANs also use the same design as in DCGANs.The $D$ network in SGANs inherits the same network topology of the pre-trained classifier $C_{real}$ and the only difference is that an additional class $N+1$ has

| Metric | | CIFAR-100 | Flower-102 | MS-Celeb-1M |
|---|---|---|---|---|
| $C_{real}$ | | 69.1% | 97.6% | 99.5% |
| $C_{GAN}$ | CGANs | 9.4% | 13.7% | 16.8% |
| | GANs | 36.8% | 48.6% | 56.8% |
| | WGANs | 44.1% | 59.1% | 64.7% |
| | SGANs | 49.6% | 61.8% | 68.7% |
| **GQI** | CGANs | 17 | 14 | 17 |
| | GANs | 53 | 50 | 57 |
| | WGANs | 64 | 61 | 65 |
| | SGANs | 72 | 63 | 69 |
| Inception Score (Mean/Std) | CGANs | 3.24 / 0.52 | 1.73 / 0.41 | 0.18 / 0.01 |
| | GANs | 12.41/1.38 | 9.84/1.17 | 1.23 / 0.03 |
| | WGANs | 14.86 / 2.17 | 11.37 / 1.85 | 1.61 / 0.02 |
| | SGANs | 16.04 / 2.68 | 11.94 / 1.92 | 1.79 / 0.04 |

Table 1: $GQI$ of various GANs on the three datasets: CIFAR-100, Flower-102, and MS-Celeb-1M. At the second row are the Top-1 accuracies of the $C_{real}$. The section corresponding to $C_{GAN}$ are the Top-1 accuracies of $C_{GAN}$. The section corresponding to $GQI$ are the GAN Quality Indices. The bottom four rows are the inception scores.

| | N=0 | N=1 | N=10 | N=50 | N=100 |
|---|---|---|---|---|---|
| **GQI** | 69 | 68 | 66 | 65 | 64 |
| Inception Score | 1.79 / 0.04 | 1.80 / 0.04 | 1.79 / 0.04 | 1.78 / 0.04 | 1.79 / 0.04 |

Table 2: GQI vs. Inception Score on evaluating SGANs after removing generated images of $N$ randomly selected classes on MS-Celeb-1M dataset.

been added in the output softmax layer. The $D$ weights in SGANs are initialized with a Normal distribution and trained from scratch together with the Generator. In all the GAN model training, no pre-processing is applied to training images besides scaling to the range of $[-1, 1]$. All models were trained with mini-batch stochastic gradient descent (SGD) with a mini-batch size of 256. All weights are initialized from a zero-centered Normal distribution with standard deviation 0.02. The learning rate is set to 0.0002 and the momentum is 0.9 through all the training.

For GANs evaluation, a classifier $C_{GAN}$ is trained with the generated data for each GANs model. The training for each $C_{GAN}$ follows the same network design and parameter setting as the corresponding $C_{real}$.

**Results:** Table 1 shows the $GQI$ of 4 different types of GANs on the three datasets. The Top-1 accuracies of the classifiers $C_{real}$ trained on real data are shown at the second row. The accuracies of the GAN-induced classifers are shown on the next four rows. The bottom four rows are the GAN Quality Indices. It is interesting to note that the $GQIs$ for each type of GANs are quite consistent across the three datasets. Among the four different types of GANs, CGANs is the worst. The $GQIs$ on the three datasets are in the range of 10 to 20. SGANs is the best, and the $GQIs$ are in the range of 60 to 70.

We have computed the inception scores on the three datasets for comparison. The results for the four different GANs are shown in Table 1 (the bottom four rows). We can see that the inception scores are consistent with the $GQI$ indices in terms of ranking. It is important to note that inception score cannot replace the classification based metric. As pointed out by Che et al. (2017)) and Hendrycks & Basart (2017), a model could attain a high inception score even if it produces a single compelling image and does not capture the dataset diversity. Inception score measures whether varied images are generated and whether the generated images contain meaningful objects, but it does not take into account the distribution of the class labels in the real dataset. In fact, as shown in Table 2, if we remove all the images from one or more randomly selected classes, the inception score of the remaining images changes very little, yet the $GQI$ decreases noticeably.

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
