# OpenReview forum: "GAN Quality Index (GQI) By GAN-induced Classifier"
_ICLR.cc/2018/Workshop — Reject_

### Official Review · AnonReviewer3 · 2018-02-22

**Rating:** 6
**Confidence:** 4

**Review:**

The authors propose a new automated method for assessing the quality of images from a generative model. In particular, the authors propose building a classifier on the generated data and then testing the new classifier on the original/real data.

The authors show that this metric (GQI) provides sensible preliminary results that allows a researcher to discriminate between the quality of various GANs. The paper is well written with clear explanations and easy-to-understand experiments.

Pros:
- Simple and easy to understand metric for assessing synthetic image quality.
- Results are reasonable.
- Clear explanations.

Cons:
- The field is full of image quality metric for GANs: inception score, frechet distance, etc. It is not clear which metric is best and this paper makes no argument that their metric is superior overall.
- The sensitivity of this metric to measure diversity in a generated dataset is not weak (Table 2 ranges from 64-69).
- Having a single metric that measures everything is difficult to work with because it conflates single sample quality with the diversity across images in a distribution. That is, it is not clear how the two *separate* issues interplay in the overall GQI score.

---

### Official Review · AnonReviewer1 · 2018-03-08
**Ok but not good enough**

**Rating:** 4
**Confidence:** 4

**Review:**

The paper introduces a metric called GQI to evaluate GAN models. It evaluates the accuracy of a GAN-induced classifier on real images, which can be view as a "symmetric version" of the inception score.

The goodness of the GQI is that it considers the diversity of the generated samples, compared with inception score. However, personally, I think the core problem of the inception score metric is that it cannot measure the generalization ability of GAN models. In particular, simply memorizing the training samples can give the almost highest score. The same problem exists for the proposed GQI method.

Overall, the contribution of the paper is limited and it doesn't reach the acceptance threshold.

---

### Official Review · AnonReviewer2 · 2018-03-10
**Not enough literature review, limited novelty.**

**Rating:** 5
**Confidence:** 5

**Review:**

[overview]

This paper proposed a new metric called GAN Quality Index (GQI) for evaluating the performance of different GAN models. In this metric, the authors used GAN-generated samples to train a classifier and then apply it on real test set. The classification accuracy on real test set is regarded as a good signal to indicate the performance of generators.

[comment]

This paper presented a new metric GQI. It seems reasonable. However, this kind of metric has been proposed by previous paper:

LR-GAN: Layered Recursive Generative Adversarial Networks for Image Generation
https://arxiv.org/pdf/1703.01560.pdf

In the above paper,  the authors proposed to use adversarial accuracy and adversarial divergence evaluate the generators, which also trained a classifier based on the generated samples. The difference is that they trained generators for all categories separately, and thus could directly generate categorical samples, instead of using another classifier to classify the generated samples. I think this way is more reasonable than the one used in this paper.

Considering the above reference, I think the novelty in this paper discounts much. It would be nice for the authors to make more comprehensive comparison with previous evaluation metrics scattered in different papers. Another reference here:

Pros and Cons of GAN Evaluation Measures
https://arxiv.org/pdf/1802.03446.pdf

---

### Decision · Program_Chairs · 2018-03-20
**ICLR 2018 Workshop Acceptance Decision**

**Decision:**

Reject

**Comment:**

Based on the reviews, this paper has not been accepted for presentation at the ICLR workshop. However, the conversation and updates can continue to appear here on OpenReview.